# Enhancement of solute clearance using pulsatile push-pull dialysate flow for the Quanta SC+: A novel clinic-to-home haemodialysis system

**Clive Buckberry**[1]☯*, **Nicholas Hoenich**[2]☯, **Detlef Krieter**[3], **Horst-Dieter Lemke**[4]‡, **Marieke Rüth**[4]‡, **John E. Milad**[1]☯

**1** Quanta Dialysis Technologies Ltd, Alcester, Warwickshire, United Kingdom, **2** Newcastle University, Newcastle upon Tyne, United Kingdom, **3** University of Wurzburg, Wurzburg, Germany, **4** EXcorLab GmbH, Industrie Center Obernburg, Obernburg, Germany

☯ These authors contributed equally to this work.
‡ These authors also contributed equally to this work.
* clive.buckberry@quantadt.com

**Data Availability Statement:** All relevant data within the manuscript and its supporting information files is available from the authors and

## Abstract

### Background and objective

The SC+ haemodialysis system developed by Quanta Dialysis Technologies is a small, easy-to-use dialysis system designed to improve patient access to self-care and home haemodialysis. A prototype variant of the standard SC+ device with a modified fluidic management system generating a pulsatile push-pull dialysate flow through the dialyser during use has been developed for evaluation. It was hypothesized that, as a consequence of the pulsatile push-pull flow through the dialyser, the boundary layers at the membrane surface would be disrupted, thereby enhancing solute transport across the membrane, modifying protein fouling and maintaining the surface area available for mass and fluid transport throughout the whole treatment, leading to solute transport (clearance) enhancement compared to normal haemodialysis (HD) operation.

### Methods

The pumping action of the SC+ system was modified by altering the sequence and timings of the valves and pumps associated with the flow balancing chambers that push and pull dialysis fluid to and from the dialyser. Using this unique prototype device, solute clearance performance was assessed across a range of molecular weights in two related series of laboratory bench studies. The first measured dialysis fluid moving across the dialyser membrane using ultrasonic flowmeters to establish the validity of the approach; solute clearance was subsequently measured using fluorescently tagged dextran molecules as surrogates for uraemic toxins. The second study used human blood doped with uraemic toxins collected from the spent dialysate of dialysis patients to quantify solute transport. In both, the performance of the SC+ prototype was assessed alongside reference devices operating in HD and pre-dilution haemodiafiltration (HDF) modes.

has been uploaded onto PLos One as supporting information as excel spreadsheets.

**Funding:** This work was funded by Quanta Dialysis Technologies as part of its future research programme of engineering development. That included funding of eXcorlab by Quanta to provide an independent assessment using another manufacturers dialysis machine. The funder provided support in the form of salaries and facilities for authors CB, JM who are full-time employees of the funder. eXcorlab were contracted as an independent company to Quanta in provide objective evidence against a known industry standard. NH and DK are independent consultants and members of each companies own Medical Advisory Board respectively. It was the author CB, who determined the study design and developed the experimental test methods at Quanta. eXcorlab protocols were developed independently of Quanta's in order to maintain objectivity by the authors HDL and MR.

**Competing interests:** Our commercial affiliation does not alter our adherence of any author to all PLOS ONE policies. Please note that this development is not a commercially available machine so there can be no financial gain and is considered by ourselves as precompetitive research to be shared. From the perspective of intellectual property, please note methods of treatment are not patentable under European law so we do not have competing interests in that regard either.

## Results

Initial testing with fluorescein-tagged dextran molecules (0.3 kDa, 4 kDa, 10 kDa and 20 kDa) established the validity of the experimental pulsatile push-pull operation in the SC+ system to enhance clearance and demonstrated a 10 to 15% improvement above the current HD mode used in clinic today. The magnitude of the observed enhancement compared favourably with that achieved using pre-dilution HDF with a substitution fluid flow rate of 60 mL/min (equivalent to a substitution volume of 14.4 L in a 4-hour session) with the same dialyser and marker molecules.

Additional testing using human blood indicated a comparable performance to pre-dilution HDF; however, in contrast with HDF, which demonstrated a gradual decrease in solute removal, the clearance values using the pulsatile push-pull method on the SC+ system were maintained over the entire duration of treatment. Overall albumin losses were not different.

## Conclusions

Results obtained using an experimental pulsatile push-pull dialysis flow configuration with an aqueous blood analogue and human blood ex vivo demonstrate an enhancement of solute transport across the dialyser membrane. The level of enhancement makes this approach comparable with that achieved using pre-dilution HDF with a substitution fluid flow rate of 60 mL/min (equivalent to a substitution volume of 14.4 L in a 4-hour session). The observed enhancement of solute transport is attributed to the disruption of the boundary layers at the fluid-membrane interface which, when used with blood, minimizes protein fouling and maintains the surface area.

## Introduction

In patients with end-stage renal disease kidney function must be replaced by artificial means (i.e. haemodialysis or peritoneal dialysis) or by transplantation in order to sustain life. The most commonly used treatment is haemodialysis, a process that involves the use of an artificial semi-permeable membrane. During haemodialysis, abnormal patient biochemistry is normalized primarily by diffusion; the fluid gained between treatments is removed by a hydrostatic pressure gradient across the dialyser membrane, a process referred to as ultrafiltration.

It is estimated that the number of patients receiving renal replacement therapy globally will increase to 4.9 million by 2025 [1]. The majority of patients receiving haemodialysis do so as outpatients in standalone or facility-based dialysis units, typically three times per week for a minimum of four hours each time. The regimen of kidney replacement therapy is associated with poor health outcomes, can be burdensome for patients and their support networks, and is costly for healthcare payers [2].

Haemodialysis offered in the home setting (HHD) is a more cost-effective treatment option in the long term [3]. It also provides patients with the ability to dialyse on a flexible schedule more frequently and/or for longer periods. Importantly, when patients are treated in their own home, they have lower rates of dialysis-related complications, hospitalisations and mortality [4–6]. Moreover, HHD also provides patients with quality-of-life improvements [7].

Despite these advantages, uptake of HHD has been low. Several factors have been implicated [8], most notably the fear associated with self-managing haemodialysis treatments at home, which can be a significant barrier when deciding on modality type [9,10].

Traditionally, HHD has been performed with machines identical to those used in a hospital setting. Such machines are typically large, cumbersome, intimidating and intrusive in the home setting. Although some have been adapted for home use, more recently manufacturers have begun focusing on machines specifically developed for home therapy use and patient operation [11, 12]. Such an approach has necessitated an improved understanding of industrial design, human factors and ergonomics to ensure that the burden of undertaking treatment in the home is reduced for the patient and their care partner [13, 14].

It is desirable for haemodialysis machines to not only be suitable for home use, but also to be suitable for use in dialysis facilities, thereby allowing dialysis programmes to balance and optimize clinical resources and to transition patients from one treatment setting to another using a single platform across the continuum of care. It is with this in mind that the Quanta SC+ personal haemodialysis system was developed.

The process of haemodialysis favours the removal of low molecular weight solutes. The removal of middle or high molecular weight solutes is limited by the characteristics of the dialysis membrane or artificial kidney. Limited enhancement of small molecule removal can be achieved by modifying diffusive forces through changes in dialyser design [15, 16], but enhancing the removal of middle or larger molecules requires a different approach [17]. Briefly, such an approach involves the combination of diffusion with convection [18]. This combination may be achieved either by internal filtration [19], haemodiafiltration (HDF) [20], or by the use of a new generation of membranes, such as medium cut-off membranes, in a conventional dialysis setting [21–24]. One of the unintended consequences of these approaches is enhanced protein removal [25].

The use of on line haemodiafiltration (olHDF), defined as a blood purification therapy combining diffusive and convective solute transport using a high-flux membrane characterized by an ultrafiltration coefficient greater than 20 mL/h/mmHg/m$^2$ and a sieving coefficient (S) for β2-microglobulin of greater than 0.6, in which the fluid balance is maintained by external infusion of a sterile, non-pyrogenic solution into the patient's blood, is becoming common, with more than 100,000 patients in Europe and Japan being treated by such an approach [20, 26].

olHDF offers a number of advantages compared to haemodialysis [27]; however, the technique does not fit well with the simplified use and time flexibility associated with HHD. On the other hand, expanded dialysis using dialysers utilizing medium cut off membranes requires nothing more than a change in dialyser [21–23].

To improve HHD adoption rates and procure the associated benefits, Quanta Dialysis Technologies Ltd (Alcester, UK) has adopted a new approach to haemodialysis system design which encourages patients to safely take control of their own treatments within the home setting (Fig 1). Many design elements of the SC+ system are recognisable in other clinically used dialysis machines but in the SC+ there are a number of key differences, namely that all the elements of the dialysis fluid pathway have been placed onto a disposable cartridge (Fig 2).

In this paper, a potential future development of the SC+ platform that could enhance solute clearance at higher molecular weights, while remaining true to its design intent of simplicity without compromising standards of care is, described and applied.

## Operating principle of the SC+ haemodialysis system

During operation of the SC+ haemodialysis system, the dialysis fluid flows through the cartridge in discrete packets (Fig 3). This is achieved by the application of pneumatic pressure and

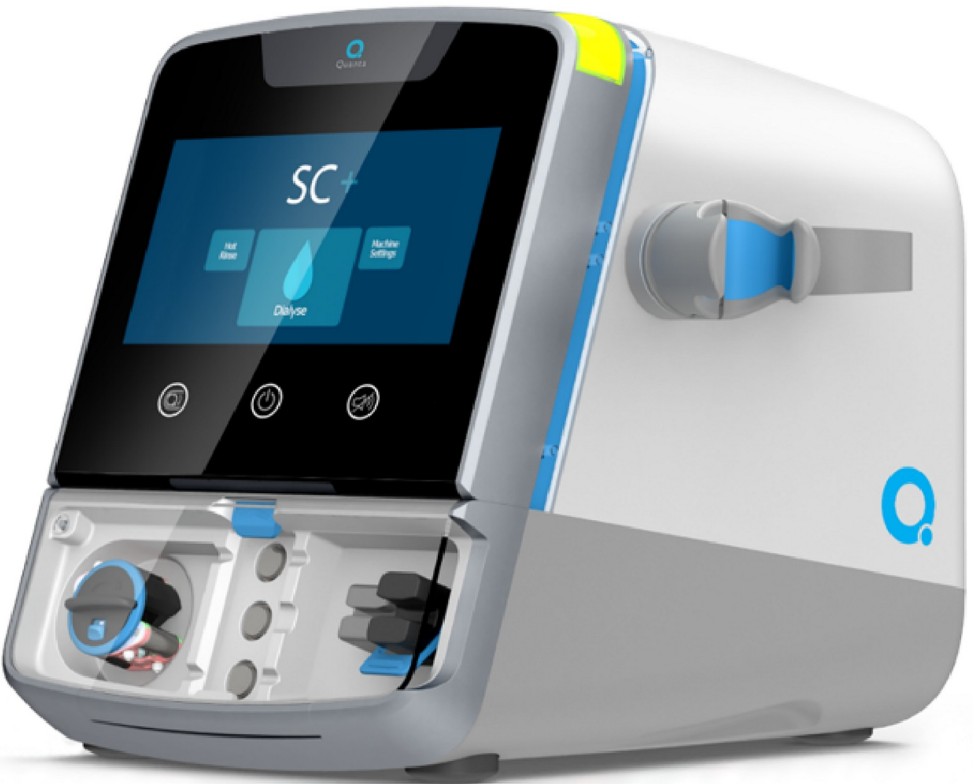

**Fig 1. Quanta SC+ haemodialysis system.**

vacuum to manipulate a flexible PVC membrane that, in turn, opens and closes a sequential series of valves and pump cavities or flow balance chambers. The actuation of the membrane at each valve and cavity is independently operated by a uniquely addressable solenoid valve. During normal operation, the two flow balance chambers are operated "in phase", such that they simultaneously move fluid to and from the dialyser in 22 mL packets, drawing fresh dialysis fluid into the system and expelling used dialysis fluid during each half cycle. In the current CE certified design used clinically, this sequence is hard coded into the operating software.

This study investigated an experimental arrangement whereby the pulsed flow into and out of the dialyser was altered by changing the relative phase of both the valves and pumps associated with each balance chamber. Using this approach, the flow can be either rapidly accelerated through the dialyser or forced across the fibre walls from the dialysate to the blood side and back again (Fig 3). Such an approach enhances the distribution of fresh dialysis fluid around the fibres and disrupts the boundary layer surrounding them, thereby enhancing solute transport.

This approach builds on the concept of push-pull in haemodialysis, which has demonstrated enhanced solute clearances compared to conventional haemodialysis without any increased loss of albumin [28–31].

## Materials and methods

A stepwise approach was used whereby a proof of concept study was performed initially to demonstrate that the software changes to the pump and valve sequencing produced a significant shift in fluid across the dialyser membrane within the time period available for normal

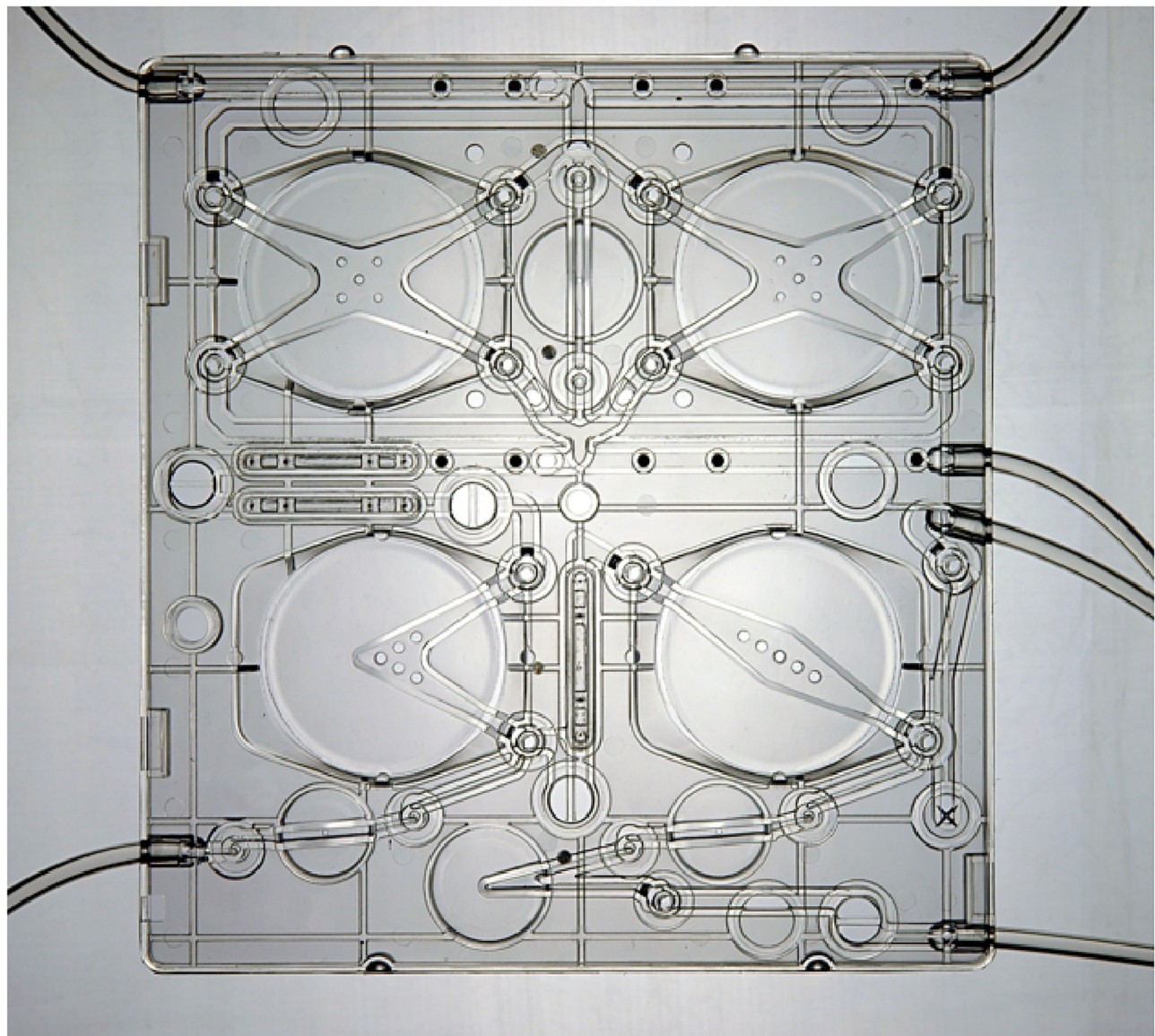

**Fig 2. SC+ dialysate cartridge.**

operation at a dialysate flow rate, $Q_d$, of 500 mL/min. Then, using a series of representative molecular markers, the alteration of clearance was quantified and used to form an initial understanding of the dynamics of this development, which would steer future refinements. Finally, the clearance characteristics of the dialysis system using a non-HDF-specific dialyser (Leoceed 18H - Asahi Kasei Medical Co. Ltd, Dusseldorf, Germany) were measured ex vivo with human blood against a benchmark machine used in current clinical practice.

All donor blood was received as reagents for *ex vivo* experiments. Therefore, the human donors did not participate directly in the study. Also, each donor gave his informed consent in writing to a detailed description of what was planned with the donated blood prior to the experiments. This document was written in accordance with the General Data Protection Regulation (GDPR) of the EU. Prospective ethics approval for this study was obtained through members of Quanta's medical advisory board.

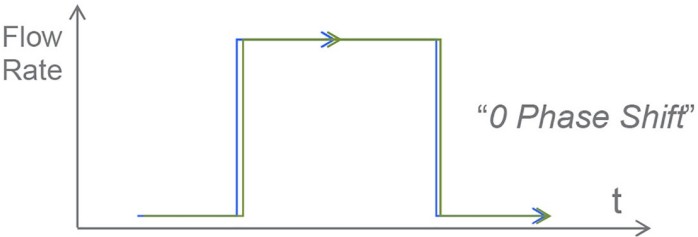

(a)        Standard Operating mode

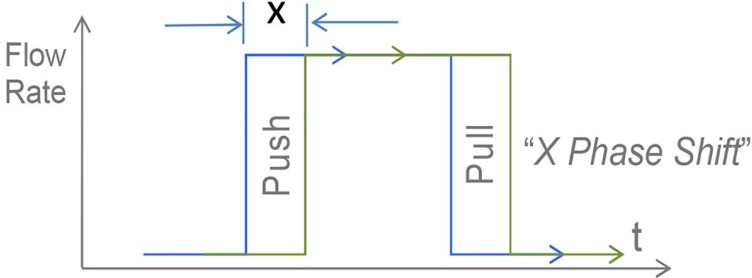

(b)        Operating mode with pulse flow generated by a phase shift

**Fig 3. Phase varied flow balance pump sequencing.**

## Proof of concept

To demonstrate the proof of concept a dual channel ultrasonic transducer system was attached to the dialysis fluid lines leading to and from the dialyser and used to measure the instantaneous fluid velocity between the flow balance chambers and the dialyser. Simultaneously, the fluctuations in the mass moved across the dialyser were determined gravimetrically and correlated with the transducer data. The blood flow rate, $Q_b$, was maintained at 300 mL/min during the studies to represent a patient whose vascular access cannot deliver the high blood flow rates necessary for HDF therapies.

## Solute transport or clearance

Two series of experimental studies to quantify solute transport or clearance were performed. The first series of studies was undertaken at Quanta Dialysis Technologies laboratories using fluorescein-labelled dextrans [32]. A second series of studies was performed independently at eXcorLab GmbH (Obernburg, Germany) in accordance with methods detailed in ISO 8637–1:2017 Extracorporeal systems for blood purification—Part 1: Haemodialysers, haemodiafilters, haemofilters and haemoconcentrators. In this series of studies, the solute clearances of the push-pull mode of the SC+ haemodialysis system and the pre-dilution HDF mode of the Nikkiso DBB-05 system were compared, using the Leoceed 18H dialyser in conjunction with aqueous solution and human blood. The pre-dilution mode was chosen as it is known in the

**Table 1. Key uraemic molecules and the analogue equivalents used.**

| Uraemic Toxin (Molecular Weight, Da) | Analogue Equivalent (Molecular Weight, Da) |
| --- | --- |
| Urea (60) | Sodium chloride (58.8) |
| Inulin (522) | Fluorescein (330) |
| Vitamin B12 (1355) | Dextran + Fluorescein (4000) |
| β2-microglobulin (11000) | Dextran + Fluorescein (10000) |
| Myoglobin (16700) | Dextran + Fluorescein (20000) |
| Immunoglobulin LC (28000–56000) | Dextran + Fluorescein (40000) |

art to be more like push-pull [33]. In addition, aqueous clearance measurements were established with the Nikkiso DBB-05 system in haemodialysis mode (HD mode) to validate the experimental methods and compare the data generated with those given by the dialyser manufacturer.

**Establishment of solute clearance using fluorescein-labelled dextrans.** The focus in these experiments was more on the relative performance of differing modalities than on absolute clearance. The clearance achieved by the novel push-pull mode of the SC+ system was quantified using molecular analogues for uraemic toxins [32,34] (Table 1). Because of their cost effectiveness, their long association with the measurement of solute transport in membranes used for dialysis and their ready availability across a wide range of molecular weights (4 to 70 kDa), fluorescein-tagged dextrans were chosen and added to dialysis fluid to form a solution with a total conductivity of 14 mS/cm. Fluorescein concentrations were measured using an Aquafluor handheld fluorometer (Turner Designs, San Jose, Ca, USA). Diluted stock solutions were used to assess errors due to variations in temperature, cuvette variation/placement and dialysis fluid composition. The lowest level of detection was 0.2 ppb with a dynamic range of 3 orders of magnitude at a wavelength of 515 nm.

Before performing the clearance studies, a series of separate studies was performed to confirm that fluorescein-tagged dextran was not adsorbed by either the PVC tubing used within the extracorporeal tube sets of the SC+ device or by the polysulfone fibres of the dialysers used.

The suitability of other markers for smaller molecules in the region of 0.5 kDa to 1.5 kDa was also assessed; however, results demonstrated an unacceptable variability due to their solubility, absorbance to fibres in the dialyser and lack of optical sensitivity. In view of this, fluorescein (which has a molecular weight of 0.33 kDa) was used on its own, with concentrations measured as above. For urea, sodium chloride was used as a proxy, with concentrations measured using a temperature-compensated conductivity meter.

Two approaches were used, one in which the fluid containing the marker solutes flowed directly to waste (single pass) while in the other the fluid was recirculated. The selection of single pass or recirculation was based on the practicality of swapping between different molecular weights and the time taken to stabilize operating conditions. The former was used with small molecular weights, while the latter was used for middle and large molecular weight solutes [34].

A schematic of the flow circuit is shown in Fig 4. Temperature-controlled water baths were used to stabilize the temperature at 37˚C. The "patient" reservoir was suspended from scales to monitor the maintenance of flow balance. Flow rates were calibrated before each run. All machine alarms were enabled except the transmembrane and venous pressure alarms. The flow circuit of the SC+ device (omitted for clarity) provided all the pumping and auto-priming with dialysis fluid for both fluid paths either side of the dialyser.

All experiments were allowed to stabilize for 30 minutes in the conventional HD mode before opening valves A and/or C to initiate the clearance testing procedure. Depending on the setting of valves A, B and C, the test solution was either recirculated or passed a single time

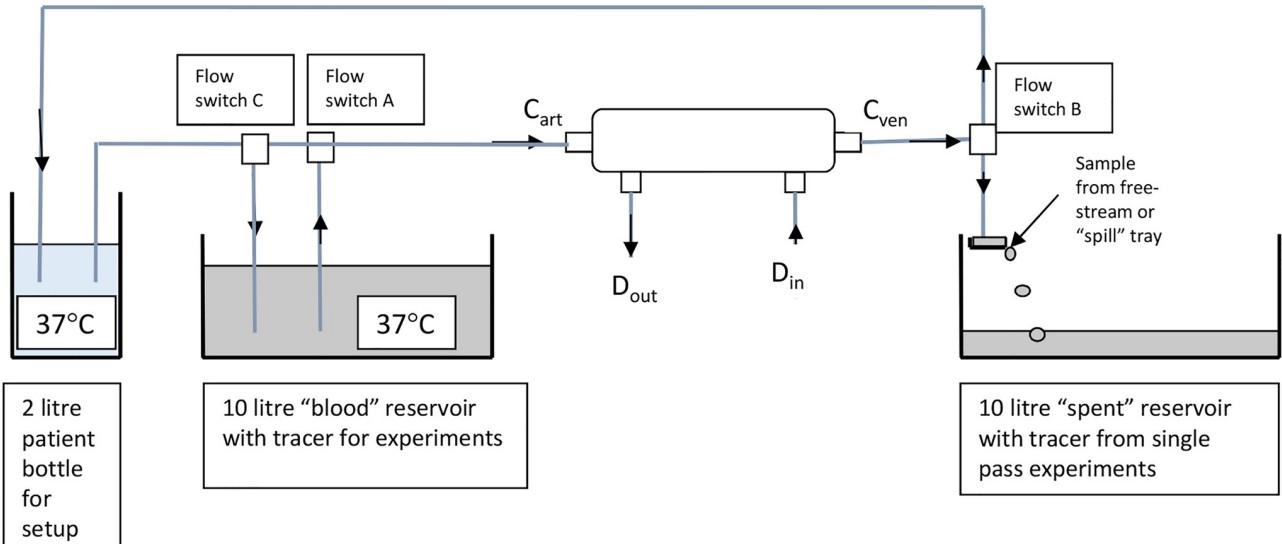

**Fig 4. Experimental flow circuit for measurement of clearance using either single pass or recirculation.**

through the dialyser. All samples were collected at valve B and then transferred to cuvettes for stabilization to room temperature and measurement.

In the single pass mode, three samples were taken at 10, 15 and 20 minutes, following stabilization for each molecular weight. When the molecular marker was changed, a 10-minute washout period preceded the stabilization period before sampling.

In the recirculation mode, single samples were taken for fluorescence measurement at 0, 3, 6 and 10 minutes following the initiation of the experiment. Samples were then taken at 5- or 10-minute intervals up to 60 or 120 minutes, depending on the molecular weight. To correct for changes due to ultrafiltration, the recirculating volume was compensated for fluid loss between sampling times.

The aqueous solute clearance (K) expressed in mL/min was calculated using Eq (1) below:

$$K = Q_b((c_{art} - c_{ven})/c_{art}) + Q_{uf}(c_{ven}/c_{art}) \tag{1}$$

in which $Q_b$ is the blood flow rate (mL/min), $Q_{uf}$ is the ultrafiltration rate (mL/min) and $C_{ven}$ and $C_{art}$ are the venous (dialyser outlet) and arterial (dialyser inlet) solute concentrations, respectively. During the experiments, the ultrafiltration rate, $Q_{uf}$, was maintained at zero and observed at all times by gravimetric flow balance measurements in both single and recirculation modes.

**Aqueous solute clearance.** Aqueous test solutions (pH 7.4 ± 0.1) composed of urea (1500 mg/L, MW 58 Da), sodium chloride (9000 mg/L, MW 58 Da), creatinine (80 mg/L, MW 113 Da), sodium dihydrogenphosphate x 2 H2O (50 mg/L, MW 156 Da), di-sodium hydrogen-phosphate x 2 H2O (230 mg/L, MW 178 Da) and inulin (Sigma, Steinheim, # 57614) (125 mg/L, MW 5200 Da) were used.

Dialysers were studied during conventional HD (target $Q_b$ = 300 mL/min; target $Q_{uf}$ = 0 mL/min; target $Q_d$ = 500 mL/min) and pre-dilution HDF (target $Q_b$ = 300 mL/min; target $Q_{uf}$ = 0 mL/min; target $Q_{sub}$ = 60 mL/min; target $Q_d$ = 500 mL/min) conditions in vitro at 37°C using the Nikkiso DBB-05 system. For the Quanta SC+ haemodialysis system, the dialysers were studied in the pulsatile push-pull mode at a $Q_b$ = 300 mL/min; target $Q_{uf}$ = 0 mL/min; programmed effective target $Q_{sub}$ = 60 mL/min at 37°C.

Before the introduction of the test solution, each dialyser was rinsed with saline by the dialysis systems in accordance with the manufacturer's instructions for use. This was followed by the priming of the extracorporeal circuit using 0.9% saline after which the test solution was introduced and approximately 1.5x the extracorporeal volume was discharged to drain before the commencement of each study. The conditions were checked for stability and samples were drawn simultaneously after 20 minutes from both the blood inlet and outlet to the dialyser, from which the urea, creatinine and phosphate concentrations were determined using a Cobas C111 clinical analyser (Roche Diagnostics GmbH, Mannheim). Inulin concentrations were determined after hydrolysis, using a commercially available assay kit (R-Biopharm, Darmstadt, Germany) using a spectrophotometer (UV-1650PC, Shimadzu Deutschland GmbH, Duisburg, Germany). Solute clearances were calculated in accordance with Eq (1).

**Plasma water clearance.**   For these determinations, heparinised (5 U/mL heparin) whole blood donated by two healthy donors was pooled and adjusted at the start of each experiment to reach a haematocrit of $32 \pm 3\%$ (actual range: 31.9–32.5%) and a total protein concentration of $60 \pm 5$ g/L (actual range: 55.4–61.8 g/L).

For the Nikkiso DBB-05 system, the study was performed using pre-dilution HDF (target $Q_b = 300$ mL/min; target $Q_{uf} = 0$ mL/min; target $Q_{sub} = 60$ mL/min; target $Q_d = 500$ mL/min) conditions ex vivo at 37°C. For the Quanta SC+ haemodialysis system, the study was performed using the pulsatile push-pull mode ($Q_b = 300$ mL/min; target $Q_{uf} = 0$ mL/min; programmed effective target $Q_{sub} = 60$ mL/min; target $Q_d = 500$ mL/min) ex vivo at 37°C.

In both series of measurements, the dialysers attached to the two dialysis systems were primed in parallel to permit simultaneous experiments.

Following priming with 0.9% saline, human blood was introduced into the circuit. Approximately 550 g of human blood (~400 mL) was used in each of the circuits. Following the introduction of blood into the circuit, conditions were allowed to equilibrate for 28 minutes. Samples were taken at 30, 32, 34 and 120, 122, 124 and 240, 242 and 244 minutes from the dialyser inlet (Cart) and outlet (Cven) and used to calculate the clearance values at 30, 120 and 240 minutes. A series of five paired experiments were performed.

Concentrated haemofiltrate containing β2-microglobulin (11.8 kDa) and myoglobin (17 kDa) extracted from the spent haemofiltrate from chronic kidney disease patients was added by infusion into the blood circuit at 28, 118 and 238 minutes before samples were drawn as described above.

To determine the albumin and total protein loss across the membrane over 240 minutes, the dialysis fluid flowing to the drain was continuously sampled at a rate of 10 mL/min, using a Ismatec IPC pump (Wertheim, Germany) linked to the dialysis fluid outflow from the dialyser.

Albumin content was established using laser nephelometry (BN ProSpec, Siemens Dade-Behring, Marburg, Germany) and total protein content determined using a Cobas C111 clinical analyser (Roche Diagnostics GmbH, Mannheim, Germany).

Throughout each experiment, the volume removed from the blood compartment for sampling was substituted by an identical volume of saline.

Plasma water clearance ($K_{PW}$) (mL/min) was calculated according to the Eq (2):

$$K_{pw} = (Q_b(1 - 0.0107[TP])(SPCHct + 1 - Hct)((c_{art} - c_{ven})/c_{art})) + Q_{uf}(c_{ven}/c_{art}) \qquad (2)$$

where Qb is the blood flow rate at the inlet of the dialyser, Quf is the ultrafiltration rate, Cven and Cart are the venous (dialyser outlet) and arterial (dialyser inlet) solute concentration, respectively, Hct is the haematocrit at the time of sampling and TP is the total protein concentration (g/L) at the same time point.

To account for solute shifts from blood cells, the solute partition coefficient (SPC) was assumed as 0 for β2-microglobulin and myoglobin.

Haematocrit was established using an ABX Pentra 60 cell counter (Agon Lab AG, Reichenbach/Stuttgart, Germany).

The weight of the blood bag was monitored throughout each experiment. An increase in the weight was noted for the SC+ system despite a target Quf = 0 mL/min, indicating the presence of back filtration (the transfer of fluid from the dialysis fluid pathway into the blood pathway). To minimize the impact of back filtration on blood composition (haematocrit and protein concentration), the pulsatile push-pull mode was occasionally interrupted.

Data analysis of plasma water clearance and albumin loss were performed by ANOVA, in which blood pool and the dialysis system (Nikkiso DBB-05 and Quanta SC+) were used as covariates, followed by pairwise comparison according to Tukey using a standard statistical package (Minitab release 17, Additive GmbH, Friedrichsdorf, Germany). A probability of $p < 0.05$ was considered significant.

## Results

### Estimation of fluid transferred during pulsatile push-pull flow using ultrasonic flow measurement

The ultrasonic flow rates in the standard (unmodified) SC+ system with zero phase delay between the opening of the two flow balance chambers in the inlet and outlet pathways at a dialysate flow rate of 500 mL/min, are shown in Fig 5. Fig 6 shows the effect of a phase delay of 300 ms. Using such a delay, the volume moved across the dialyser membrane was 8 L/hr (equivalent to 133 mL/min) in each direction. The small differences observed between the

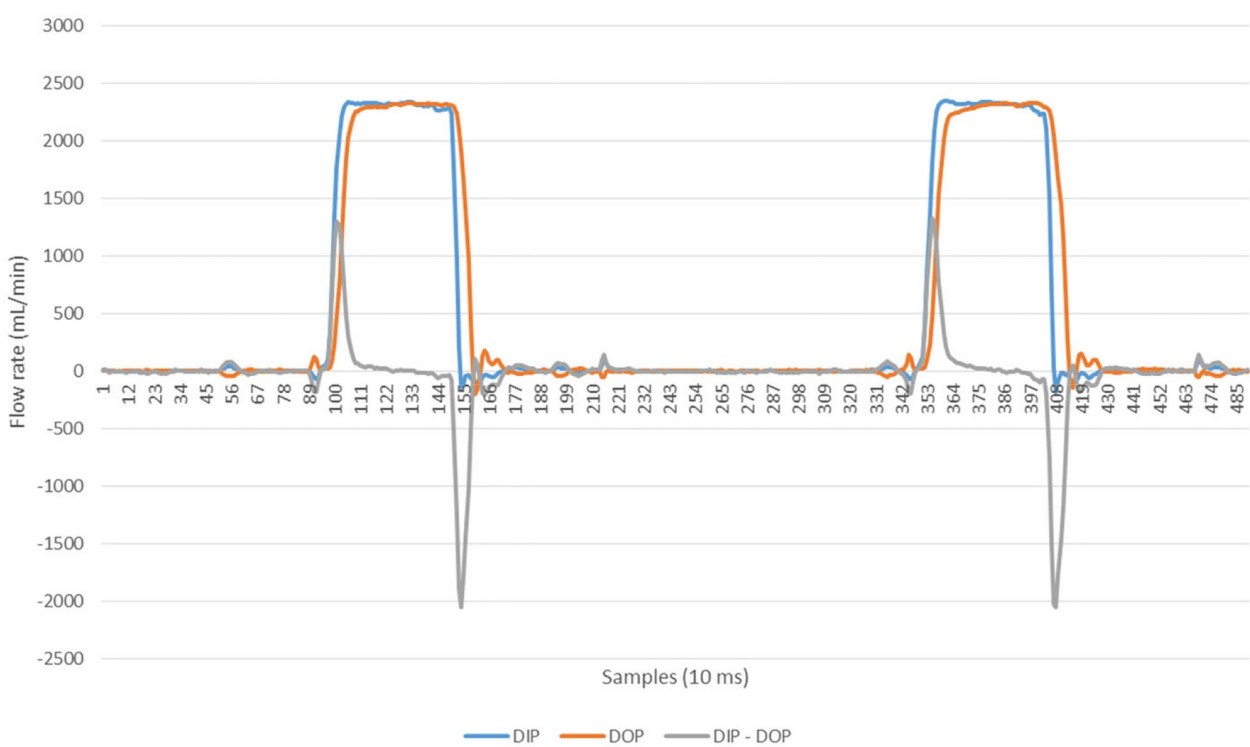

**Fig 5. Ultrasonic flow rates into and out of a dialyser with 0 ms delay between flow balance pumps.**

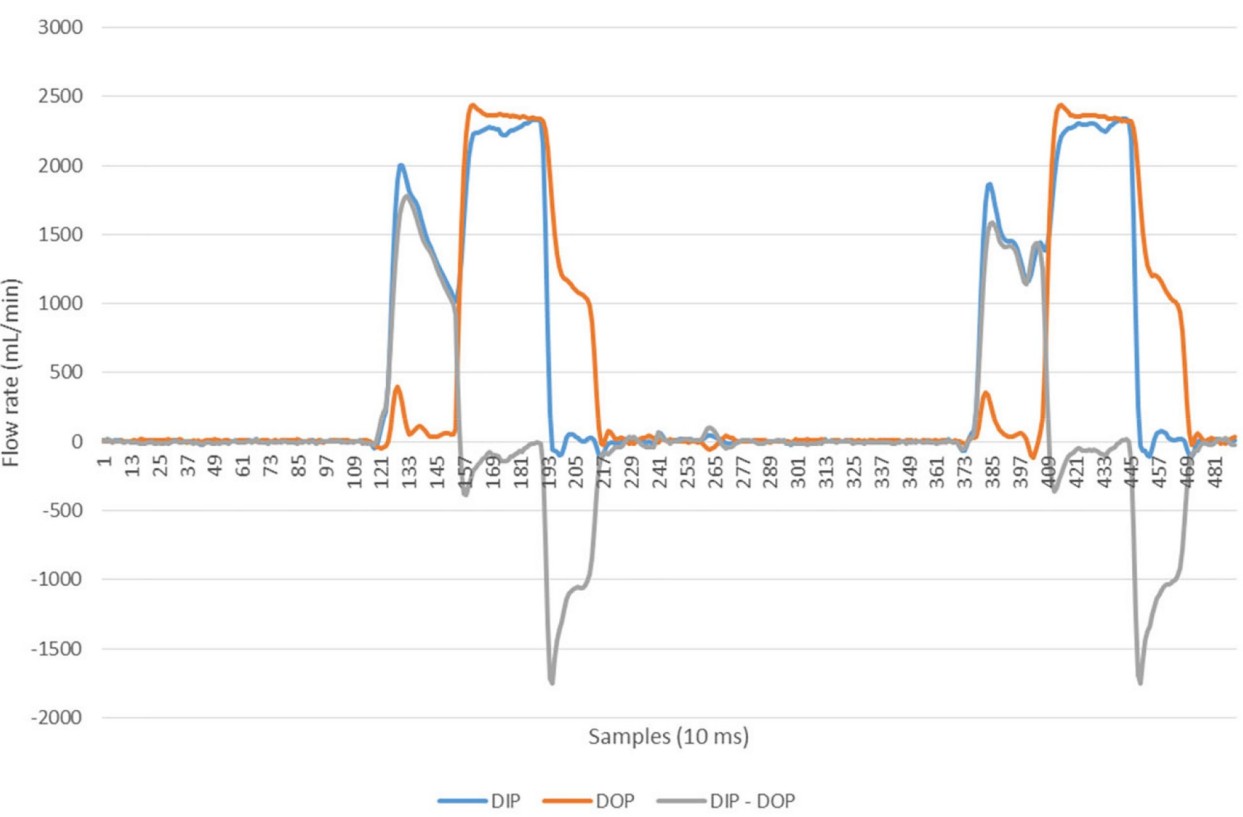

**Fig 6. Ultrasonic flow rates into and out of a dialyser with a 300 ms delay between flow balance pumps.**

push and pull portions are a result of minor differences in transmembrane pressure applied by the pneumatic pressure and vacuum cycles.

With the phase delay, large volumes of fluid could be moved across the membrane. In practice this was limited by the residence time of the bolus within the dialyser. In order to match the Nikkiso DBB-05 system when used in HDF mode, for the experimental studies described the phase delay was reduced to 200 ms (60 mL/min or 3.6 L/hr). The rationale for this reduction was twofold: first, it matched the substitution fluid infusion rate delivered when using the same dialyser in conventional on line HDF mode; second, it optimized the residence time of a bolus of fluid within the dialyser when in the pulsatile push-pull mode.

### Solute clearance using fluorescein-labelled dextrans

Having established the principle of pulsatile push-pull flow, a series of studies were performed to quantify the magnitude of enhancements that this approach delivers by using 4, 10 and 20 kDa fluorescein-labelled dextrans. The studies used the recirculating experimental set up shown in Fig 4. The use of recirculation allowed for repeated measurements to be made, mitigating any effect of collecting small samples at points that might not have been in perfect sequence with the push-pull cycle.

Data collected for each of the fluorescein-labelled dextrans are shown in Fig 7a–7c where equivalent data established during normal operation of the SC+ system are also shown.

All points shown are the average of three measurements and are shown with +/- one standard deviation error bars.

(a)

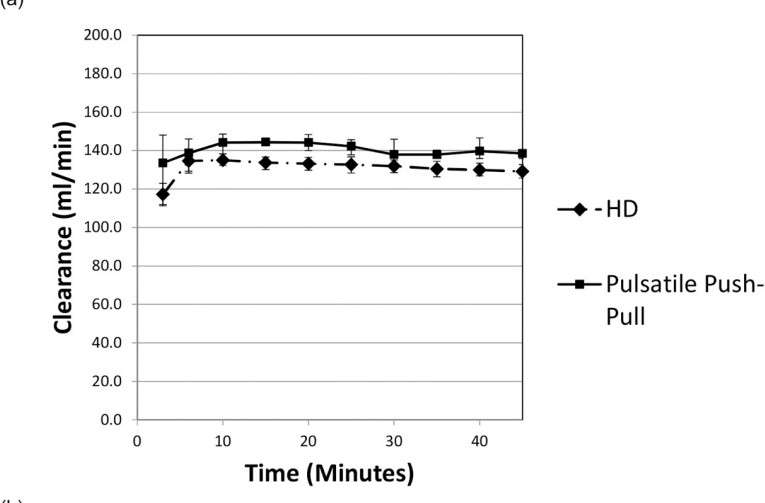

(b)

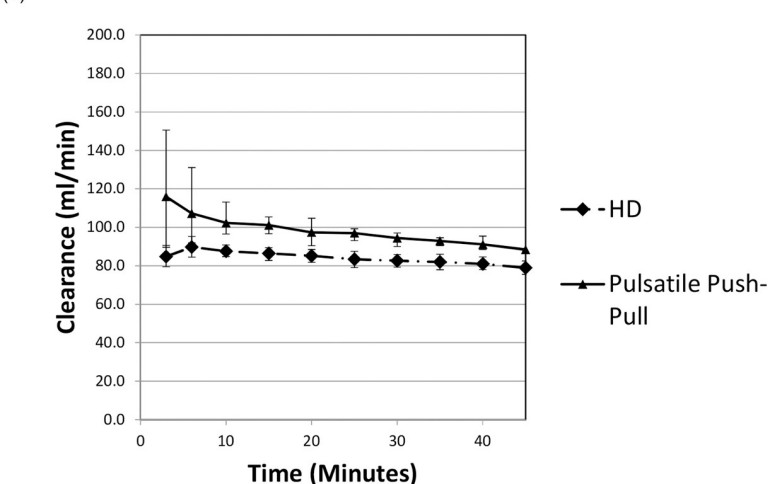

(c)

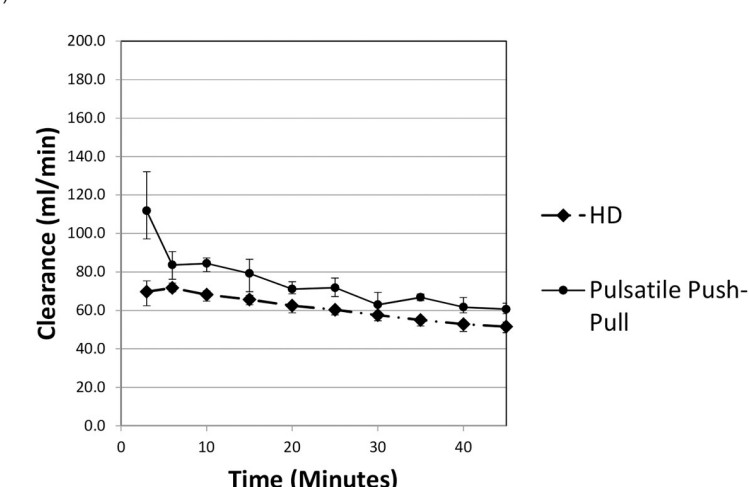

**Fig 7.** a. Variation of clearance with time using the SC+ for 4 kDa dextran in conventional flow mode and pulsatile push-pull mode. b. Variation of clearance with time using the SC+ for 10 kDa dextran in conventional flow mode and pulsatile push-pull mode. c. Variation of clearance with time using the SC+ for 20 kDa dextran in conventional flow mode and pulsatile push-pull mode.

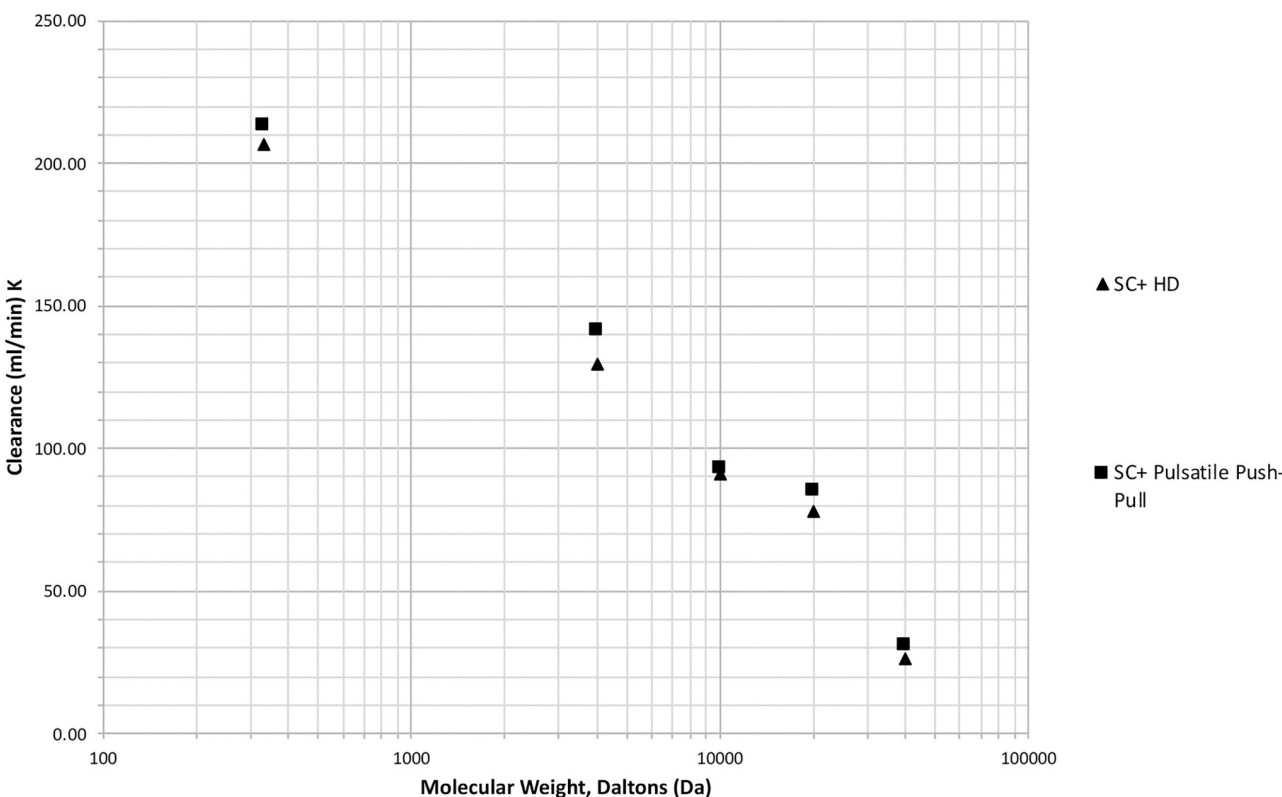

**Fig 8. Relationship between solute clearance and molecular weight in conventional flow mode and pulsatile push-pull mode with a phase delay of 200 ms for the Quanta SC+ dialysis system.**

The relationship between the observed clearances and molecular weight for the SC+ system with and without the push-pull modification using the same dialyser type (Asahi Kasei Leoceed 18H) is shown in Fig 8. The values shown are the mean values established during the individual experiments.

## Aqueous solute clearances

Aqueous solute clearance data established using urea, creatinine, phosphate and inulin, based on three experiments (N = 3) conducted at eXcorLab for the SC+ system incorporating pulsatile push-pull flow, were compared with those established using the Nikkiso DBB-05 dialysis system in conventional HD and pre-dilution HDF modes and are shown in Table 2, where the

**Table 2. Aqueous solute clearances for low molecular weights.**

| Machine | Mode | Clearance (mL/min), mean +/- SD & N = 3 | | | |
|---|---|---|---|---|---|
| | | Urea | Creatinine | Phosphate | Inulin |
| SC+ | Pulsatile Push-Pull HDF | 254±7 | 239±9 | 227±9 | 114±13 |
| Nikkiso DBB-05 | HD | 276±3 | 260±5 | 248±5 | 106±1[b] |
| Nikkiso DBB-05 | Pre-dilution HDF | 266±7 | 249±8 | 236±7 | 100±16 |
| Manufacturer's specification[a] | HD | 274±7 | 260 | 247 | n/a |

[a] Leoceed 18H Haemodialyser

All data shown are the mean of three experiments except

[b] mean of two experiments

manufacturer's product insert data for the dialyser used are also provided. The clearance measurements gathered were in broad agreement with the values specified by the manufacturer for the Leoceed 18H haemodialyser. The modified Quanta SC+ system incorporating the pulsatile push-pull flow mode yielded slightly lower clearances for urea, creatinine and phosphate compared to the Nikkiso DBB-05 system, when used in both conventional HD and pre-dilution HDF modes. The observed deviation is unlikely to be of clinical importance and was attributed to reduced diffusive forces resulting from dilution. This observation is in accordance with those of Ficheux et al [35]. For inulin, the Quanta SC+ system showed slightly higher clearances compared to the Nikkiso DBB-05 system in both HD and pre-dilution HDF modes.

## Plasma water solute clearance and albumin loss

Plasma water solute clearances measured at 30, 120 and 240 minutes are shown in Table 3. Data are presented as mean ± SD based on five experiments with three samples acquired at each time point resulting in N = 15. Overall, the modified SC+ was comparable to the Nikkiso DBB-05 system in pre-dilution HDF mode.

Comparing the different time points (30, 120 and 240 minutes), the Quanta SC+ shows constant clearances over all time points. Calculation of the area under the curve for clearance for β2-microglobulin and myoglobin indicates that there is equivalence for β2-microglobulin between the two devices. However, the SC+ shows a 16% improvement for myoglobin overall (30–240 minutes) and a 28% increase in the last two hours (120–240 minutes) of the study period.

Mean albumin loss data for the whole study period are given in Table 4 based on five experiments. SC+ with the pulsatile push-pull flow configuration showed lower albumin loss compared to the Nikkiso DBB-05 in pre-dilution HDF mode, but the difference was small and not statistically significant.

Table 3. Plasma water solute clearances for middle weight molecules.

| Machine | Mode | Sampling time (min) | Clearance (mL/min), mean +/-SD (N = 15) | |
| --- | --- | --- | --- | --- |
| | | | β2-microglobulin | Myoglobin |
| SC+ | Pulsatile Push-Pull HD | 30 | 61±10 | 35±11 |
| | | 120 | 59±7 | 26±15 |
| | | 240 | 66±5 | 30±9[a] |
| | | AUC (mL*min) | 12846 | 6098 |
| Nikkiso DBB-05 | Pre-dilution HDF | 30 | 65±9 | 32±11 |
| | | 120 | 63±10 | 26±11 |
| | | 240 | 57±10 | 18±8[a] |
| | | AUC (mL*min) | 12960 | 5250 |

[a] means are statistically significant from each other (p<0.05) at the same time point

Table 4. Albumin loss over 240 minutes.

| Machine | Mode | Albumin Loss (g), mean +/-SD (N = 5) |
| --- | --- | --- |
| SC+ | Pulsatile Push-Pull HDF | 0.80±0.49 |
| Nikkiso DBB-05 | Pre-dilution HDF | 0.88±0.40 |

All data shown are the mean of five experiments.

## Discussion

Although HDF offers a number of advantages over conventional haemodialysis, and is accepted in Europe and Japan, it is less commonly used in the United States [36]. Clinical use of HDF requires increased dialysis fluid purity, additional sterile tubing sets for fluid infusion and additional blood side pumps. This increases complexity and is at odds with self-managed treatments in terms of technology simplification or ease of use. Furthermore, solute transport during HDF is associated with membrane fouling, leading to a loss in performance, increased incidence of nuisance alarms and the need for nursing interventions [25]. Such fouling has been shown to lead to a decrease in ultrafiltration coefficient ($K_{Uf}$) of approximately 12% during HD and 16% during post-dilution HDF over a 3-hour period [35]. On the other hand, the push-pull approach does not require external substitution fluid and relies on alternate repetitions of forward and backward filtration across the dialyser membrane during dialysis treatment [37].

In the current series of studies a simple adjustment to the controlling software of the SC + system alters the sequence and timing of the valves and pumps associated with the flow balancing chambers that push and pull the dialysis fluid through the dialyser and introduces a phase shift. Ultrasonic measurements confirmed that volumes up to 8 L/hr could be easily moved across the membrane using such an approach.

Experiments with dextran yielded an increase in clearance for all molecular weights tested. In ex vivo comparative testing using the Nikkiso DBB-05 machine operating in pre-dilution HDF mode ($Q_{sub}$ = 60 mL/min), the modified SC+ produced not only an equivalent clearance, but this clearance was maintained over the full duration of the experimental treatment period (240 minutes) and was without any increase in the loss of albumin.

During dialysis, when uraemic solutes are removed from the blood through the dialyser membrane with a variable pore size distribution, concentration polarization occurs and a protein gel layer develops on the membrane surface. The concentration polarization may be viewed as an additional mass boundary layer affecting solute transport, while protein gel layer offers additional structural resistance affecting fluid transport [38,39].

The improved results observed using the SC+ system with a modified fluidic management system suggest a disruption of the boundary layers at the blood-membrane interface. This minimizes protein fouling and maintains the surface area available for mass and fluid transport throughout the whole treatment period. Further studies have been initiated to validate this hypothesis, by specifically quantifying the mitigation of the protein layer build up during the process of boundary layer disruption.

The potential functional and usability advantages of this approach are multiple: no additional sterile tubing or pumps are required to deliver a functional performance that is comparable with that achieved by HDF; there is no change in system complexity as far as the patient or user is concerned as a shift from conventional haemodialysis to pulsed push-pull haemodialysis can be achieved by the push of a button; and, because pressure fluctuations in the pulsed push-pull HD mode are lower than in HDF, there is a reduction of membrane fouling and the incidence of alarms during use is likely to be reduced. Additionally, this technique may be suitable for use over extended treatment periods, such as nocturnal haemodialysis.

All these advantages make the system more suitable for self-care or HHD applications. Importantly, this approach is also fully compatible with all current and future dialyser developments, although further work will be required to confirm that the enhanced performance observed is maintained for a range of clinically used haemodialysers.

While the approach described is relatively simple in principle, a number of technical and regulatory challenges remain. Of these, some are relatively simple to address, such as the re-tuning of venous and transmembrane pressure alarms. Others, such as maintaining an

accurate flow balance, are more challenging. The most challenging aspect is the attainment and maintenance of microbiological quality during each treatment and regulatory approval.

During HDF, the fluid infused is produced continuously on line by the passage of a fraction of the dialysis fluid through bacterial and endotoxin retentive filters validated to produce sterile and non-pyrogenic fluids.

Whereas the production of on line infusion fluid involves a treatment cascade in which multiple filters may be used [40], the current approach uses the dialyser to ensure that biological contaminants are not transferred from the dialysis fluid to the patient. This approach relies on the fact that, when using ultrapure dialysis fluid, any biological contaminants present in the fluid are retained or adsorbed within the ultrastructure of the membrane wall and do not pass into the blood stream. Clinically used polymer membranes have differing adsorptive capacities [23, 38, 41] and consequently the potential exists for the transfer of endotoxin fragments and other bacterial substances present in the dialysis fluid into the patient's blood either by convective transfer (back filtration) or by movement down the concentration gradient (back diffusion) [41, 42]. The long-term clinical relevance of this remains an unexplored aspect of haemodialysis therapy.

As the membrane contained in the dialyser itself becomes the final barrier to endotoxins and endotoxin fragments in the sterility chain/cascade, detailed risk assessments will be required to ensure that the required sterility assurance level is achieved and regulatory compliance is met. This will form the primary focus for future development of the approach described.

These studies are not without limitations. They are preliminary studies to confirm proof of concept and further studies will be required to establish optimal settings of the push-pull cycle and ultimately clinical trials in vivo. The studies were performed at two different sites, and minor differences may have been introduced by the operating settings. The duration of each study was also short, and longer studies will be required to determine stability. Future studies are planned to gather these data.

In the approach described, back filtration and ultrafiltration repeat in a relatively short time, and despite a large amount of filtration, the probability that some ultrafiltrate comes directly from dialysate back filtered during a previous phase cannot be excluded. This, in turn, will influence the results through a reduction in solute concentrations caused by dilution and may manifest as an efficiency reduction. Although the in vitro and ex vivo experiments described have shown that alternating back filtration has a positive influence on inhibiting concentration polarization and permeability reduction, further studies, in terms of pulse frequencies and stroke volumes, will be required to optimize settings.

## Supporting information

**S1 Data. Recorded results and analysis of aqueous clearance.** Mean and standard deviation error of each condition.
(XLSX)

**S2 Data. Recorded results and analysis of aqueous clearance.** Mean and standard deviation error of each condition.
(XLSX)

**S3 Data.**
(XLSX)

**S4 Data.**
(XLSX)

## Acknowledgments

The authors accept responsibility for the accuracy of data, its analysis and the writing of this report.

## Author Contributions

**Conceptualization:** Clive Buckberry.

**Data curation:** Nicholas Hoenich, Horst-Dieter Lemke, Marieke Rüth.

**Formal analysis:** Detlef Krieter, Horst-Dieter Lemke, Marieke Rüth.

**Investigation:** Clive Buckberry, Nicholas Hoenich.

**Methodology:** Clive Buckberry.

**Supervision:** Detlef Krieter, Horst-Dieter Lemke, John E. Milad.

**Validation:** Detlef Krieter, Horst-Dieter Lemke.

**Writing – original draft:** Clive Buckberry.

**Writing – review & editing:** Clive Buckberry, Nicholas Hoenich, John E. Milad.

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
