## [Decision Letter · Decision Letter 0]

26 Nov 2019

PONE-D-19-16719

Enhancement of solute clearance using pulsatile push-pull dialysate flow for the Quanta SC+: a novel clinic-to-home haemodialysis system

PLOS ONE

Dear Professor Buckberry,

Thank you for submitting your manuscript to PLOS ONE. After careful consideration, we feel that it has merit but does not fully meet PLOS ONE’s publication criteria as it currently stands. Therefore, we invite you to submit a revised version of the manuscript that addresses the points raised during the review process.

We would appreciate receiving your revised manuscript by Jan 10 2020 11:59PM. To enhance the reproducibility of your results, we recommend that if applicable you deposit your laboratory protocols in protocols.io, where a protocol can be assigned its own identifier (DOI) such that it can be cited independently in the future. For instructions see: http://journals.plos.org/plosone/s/submission-guidelines#loc-laboratory-protocols

We look forward to receiving your revised manuscript.

Kind regards,

Pasqual Barretti, Ph.D., MD

Academic Editor

PLOS ONE

Journal Requirements:

2. Please update the ethics statement in the online submission form and the methods section of the manuscript to include a statement that you obtained prospective ethics approval for this study from your medical advisory board

'The authors have declared that no competing interests exist'

We note that one or more of the authors are employed by a commercial company: Quanta Dialysis Technologies Ltd & EXcorLab GmbH.

Additional Editor Comments (if provided):

Authors must fully answer the questions of the reviewers, in particular of the reviewer 1. After my reading of the text my decision is major revision

Reviewers' comments:

Reviewer's Responses to Questions

**Comments to the Author**

1. Is the manuscript technically sound, and do the data support the conclusions?

Reviewer #1: Yes

Reviewer #2: Yes

2. Has the statistical analysis been performed appropriately and rigorously? 

Reviewer #1: No

Reviewer #2: Yes

3. Have the authors made all data underlying the findings in their manuscript fully available?

Reviewer #1: Yes

Reviewer #2: Yes

4. Is the manuscript presented in an intelligible fashion and written in standard English?

Reviewer #1: Yes

Reviewer #2: Yes

5. Review Comments to the Author

Reviewer #1: Although it is known that the Push / pull mode is advantageous for clearances because of anti-fouling, it is highly novel that the Push / pull mode is incorporated into a home hemodialysis machine and put into practical use.

・Figs. should be as clear as possible.

・Add the number of original data number and standard deviation to the table and Fig.

Reviewer #2: Buckberry CH and coworkers are submitting a study assessing the potential benefit of adding convective clearance to the SC+ hemodialysis system developed by Quanta. By modifying the time pressure algorithm of filling/emptying cassette chambers they generate a push-pull like flow. As proof of concept they developed an invitro study showing that new algorithm was able to generate this alternate push-pull flow by US. They moved subsequently to a prototype with bench testing of various solute clearances (fluorescein-tagged dextran of various molecular weight) thereafter with invitro blood simulation (conductivity as surrogate of urea, myoglobin and B2M) and compared the modified SC+ to a predilution HDF model using conventional HD and predilution HDF system (high flux filter and Nikkiso DBS5 machine). In brief, the authors showed that their modified SC+ system ensured push-pull flow increasing instantaneous clearances by 10 to 15% and compared favorably with that achieved using pre-dilution HDF with a substitution fluid flow rate of 60 ml/min with the same dialyzer and marker molecules in blood simulated experiment. Interestingly clearance values using the push-pull method on the SC+ system, were maintained over the duration of treatment.

This is an interesting concept based on invitro studies taking benefits of minimal changes and time pressure algorithm modification in an existing innovative hemodialysis device. Now, the study raises several concerns that need to be addressed for clarification and better understanding:

1. This is a proof of concept study and not a clinical validation study meaning that further clinical trials in vivo are required to confirm their findings.

2. It is not clear why the modified SC+ with push-pull flow was compared with predilution HDF since my understanding of the algorithm modification is correct, it should be better suited to be compared with postdilution HDF. Internal filtration process associated with push-pull flow regime is mimicking postdilution action and not predilution.

3. In the study design, it not clear to me why solute clearances of modified SC+ developing push-pull like flow were not compared with standard SC+ algorithm machine. In other words, purely diffusive HD versus added convective component. Such design would have been better understood for showing the added value of the new algorithm.

4. Blood based invitro study is confusing and not necessarily correct since some clearance measurements are performed in a single-pass for small molecules and others in a recirculating pass for larger molecules. It would have been preferable to perform them in the same mode based on tank recirculation with log transformation time concentration decline to obtain a true overall clearance based on the slope decline. Furthermore, there is no mass balance calculation from blood and dialysate side to validate clearance calculation.

5. Surprisingly, solute clearances obtained in predilution HDF mode with DBS5 machine are lower over the all spectrum of markers than those obtained in pure hemodialysis. This is struggling and should be explained since it is not in line with the known fact that solute clearances are higher in HDF in particular for larger molecular compounds. One can suspect something went wrong with sampling due to predilution mode or calculations or both. In this setting one can expect clearances higher by about 30 ml/min through molecular weight spectrum.

6. From a presentation and wording perspective, introduction and discussion are too long and more related to the benefits of home therapy using the SC+ device. This is not the topic of this study. The authors should stay focused on the aim of the study which is to show some benefits on solute clearances by adding a convective component with a modified algorithm. In addition, the authors referred to benefits of using internal transport phenomenons using MCO membrane in term of solute clearances in standard HD. If this is the case, why not using MCO membrane in standard HD, instead of this modified new algorithm intended to increase convective clearances? How does it compare? What are benefits and risks of combining them?

6. PLOS authors have the option to publish the peer review history of their article (what does this mean?). If published, this will include your full peer review and any attached files.

Reviewer #1: No

Reviewer #2: No

---

## [Author Response · Author response to Decision Letter 0]

15 Jan 2020

Response to reviewers PONE-D-19-16719

Comments to the Author

1. Is the manuscript technically sound, and do the data support the conclusions?

Reviewer #1: Yes

Reviewer #2: Yes

Author: Accepted

2. Has the statistical analysis been performed appropriately and rigorously? 

Reviewer #1: No

Reviewer #2: Yes

 Author: Reviewer 1 has not given any details that substantiate their statement so I will add detail as I believe to be appropriate. All the data was analysed using either Excel or ‘Minitab’ that are both recognised with for such purposes by the FDA when validated appropriately as they were. In the case of the independent external test house, excorlab, that was contracted for some of the more complex tests they were specifically chosen because of their accreditation for such work under ISO 17025 both experimentally and analytically.

3. Have the authors made all data underlying the findings in their manuscript fully available?

Reviewer #1: Yes

Reviewer #2: Yes

 Author: Accepted

4. Is the manuscript presented in an intelligible fashion and written in standard English?

Reviewer #1: Yes

Reviewer #2: Yes

 Author: Accepted

5. Review Comments to the Author

Reviewer #1: Although it is known that the Push / pull mode is advantageous for clearances because of anti-fouling, it is highly novel that the Push / pull mode is incorporated into a home hemodialysis machine and put into practical use.

・Figs. should be as clear as possible.

・Add the number of original data number and standard deviation to the table and Fig.

Author: Addressed with changes to table and text regarding sample size and SD.

Reviewer #2: Buckberry CH and coworkers are submitting a study assessing the potential benefit of adding convective clearance to the SC+ hemodialysis system developed by Quanta. By modifying the time pressure algorithm of filling/emptying cassette chambers they generate a push-pull like flow. As proof of concept they developed an invitro study showing that new algorithm was able to generate this alternate push-pull flow by US. They moved subsequently to a prototype with bench testing of various solute clearances (fluorescein-tagged dextran of various molecular weight) thereafter with invitro blood simulation (conductivity as surrogate of urea, myoglobin and B2M) and compared the modified SC+ to a predilution HDF model using conventional HD and predilution HDF system (high flux filter and Nikkiso DBS5 machine). In brief, the authors showed that their modified SC+ system ensured push-pull flow increasing instantaneous clearances by 10 to 15% and compared favorably with that achieved using pre-dilution HDF with a substitution fluid flow rate of 60 ml/min with the same dialyzer and marker molecules in blood simulated experiment. Interestingly clearance values using the push-pull method on the SC+ system, were maintained over the duration of treatment.

This is an interesting concept based on invitro studies taking benefits of minimal changes and time pressure algorithm modification in an existing innovative hemodialysis device. Now, the study raises several concerns that need to be addressed for clarification and better understanding:

1. This is a proof of concept study and not a clinical validation study meaning that further clinical trials in vivo are required to confirm their findings.

Author: The reviewer is correct and this has now been clarified in the submission

2. It is not clear why the modified SC+ with push-pull flow was compared with predilution HDF since my understanding of the algorithm modification is correct, it should be better suited to be compared with postdilution HDF. Internal filtration process associated with push-pull flow regime is mimicking postdilution action and not predilution.

Author: The reviewer makes a valid observation. In post dilution the substitution fluid will first pass to the patient before returning to the dialyser, because the fluid is added post the dialyser. In pre-dilution HDF it is added just prior to the dialyser. As a consequence, the dilution ratio is different. In push-pull the substitution fluid is added directly into the dialyser so is much more akin to pre-dilution. This likeness is also explained more fully by Tattersall ‘Online haemodiafiltration: definition, dose quantification and safety revisited’ Nephrol Dial Transplant (2013) 22: 542-550. This has been addressed and referenced in the text.

3. In the study design, it not clear to me why solute clearances of modified SC+ developing push-pull like flow were not compared with standard SC+ algorithm machine. In other words, purely diffusive HD versus added convective component. Such design would have been better understood for showing the added value of the new algorithm.

Author: Again a valid point made by the reviewer. The reasons are as follows. Firstly, in practice there are many ways to program the push-pull mechanism on the SC+ device unlike the original methods proposed in this area by Shinzato which was a simple alternating movement of fluid. In our case we vary the cycle to be one of push, push, push followed by pul,l pull, pull. When can also vary the volume in each cycle and the phase depending upon how we wish to disrupt the boundary layer around the fibres. We have chosen therefore to just make a simple comparison in order to first introduce the principle that it can be done. Secondly, we cannot also compare the baseline SC+ to any other machine because of the pulsatile nature of its dialysate flow, compared to the steady state flow of all current dialysis machine in haemodialysis. This would have required a significantly more complex suite of experiments and deconvolution in the analytical phase. 

4. Blood based invitro study is confusing and not necessarily correct since some clearance measurements are performed in a single-pass for small molecules and others in a recirculating pass for larger molecules. It would have been preferable to perform them in the same mode based on tank recirculation with log transformation time concentration decline to obtain a true overall clearance based on the slope decline. Furthermore, there is no mass balance calculation from blood and dialysate side to validate clearance calculation.

Author: Good points raised again. We found during the course of method development that single-pass and recirculation methods were complimentary. Broadly speaking markers for low molecular weights were cheap, abundant and could be monitored continuously, so single pass was used. The middle weight molecules tagged with fluorescent material we more expensive and complicated to manage to maintain constant density, in particular, hence the recirculation was more effective with constant stirring. Critically, the samples taken, both volume and timing of have to be sympathetic to the push-pull cycle to account for the dilution effect of the substitution fluid. This was done and accounted for in the flow balance calculations for net fluid removal error as prescribed under IEC 60601-2-16: 2014 in all experiments. The actual values were calculated for in Eqn2 in the Quf term. I have added a comment to this effect the text.

5. Surprisingly, solute clearances obtained in predilution HDF mode with DBS5 machine are lower over the all spectrum of markers than those obtained in pure hemodialysis. This is struggling and should be explained since it is not in line with the known fact that solute clearances are higher in HDF in particular for larger molecular compounds. One can suspect something went wrong with sampling due to predilution mode or calculations or both. In this setting one can expect clearances higher by about 30 ml/min through molecular weight spectrum.

Author: We have reviewed the analysis as prompted by the reviewers comment which is valid and well made, we had internal debate on these results too. Allow us to make our counterpoints. We can confirm that following an audit at the time we found no errors in the data collection method or analysis that formed the basis of table 2. We observe that the effect is present for both machines without bias. We also point out to the reviewer that for Inulin the SC+ in push-pull mode is greater than the Nikkiso in HD so it is not true for all spectrum markers as the Reviewer suggests. It is our observation that previous studies that compare HD with HDF modalities the dialyser used will vary. HDF therapies typically employ HDF specific dialysers which are 10% larger in surface area particularily in post-dilution HDF. So often you may see Fx80 dialysers compared to Fx800. in addition a higher dialysate flow rate to counteract the dilution is often employed, so rather than 500ml/min you will see 600ml/min or higher. This gives a bias which we have avoided and may account for the reviewers perception. 

For transparency we included the manufacturers data in HD mode to compare to the experimental data HD data on the Nikkiso and replicated this very well for urea, Creatine and Phosphate again confirming our methodology experimentally and analytically. 

What we found most interesting is the effect of blood flow rate Qb to all methods. In pre and post HDF the Qsub is added to the Qb and this amplifies the calculated rate of clearance. In push-pull as we have deployed it Qsub is not strictly present in the same way, especially as it is first negative then positive direction of flow, so whilst the calculation is there the physical action is very different. This is also different to how push-pull was employed by Shinzato. What is important in this study and it’s purpose of course, is how do the two systems compare, we therefore chose to apply the equations as they currently stand without bias. We are planning to write further papers that analytically model our embodiment of pulsatile push-pull as we felt at this stage it would detract from this early proof-of-concept.

6. From a presentation and wording perspective, introduction and discussion are too long and more related to the benefits of home therapy using the SC+ device. This is not the topic of this study. The authors should stay focused on the aim of the study which is to show some benefits on solute clearances by adding a convective component with a modified algorithm. In addition, the authors referred to benefits of using internal transport phenomenon’s using MCO membrane in term of solute clearances in standard HD. If this is the case, why not using MCO membrane in standard HD, instead of this modified new algorithm intended to increase convective clearances? How does it compare? What are benefits and risks of combining them?

Author: Again, good observations. We appreciate the opinion of the reviewer, but we feel it important to explain the context of attempting this early proof of concept prototype. The delivery of haemodialysis to patients is beginning to undergo a seismic change form clinic to home-based therapy and in writing this we felt it important for the reader to know why such a different form of HDF was being attempted, one that requires no additional tubing or modification to the extracorporeal circuit and or requirement for a ‘special’ dialyser in order to maintain simplicity and ease of use. 

Use of dialysers with added internal transport such as MCO are an equally viable alternative, but currently they are being charged at a premium like HDF. We asked Baxter for samples but were declined. In this concept enhanced convective therapies based upon software programmable machine changes only would democratise HDF for all without any additional materials and be deliverable by prescription over the internet as appropriate through a secure digital health platform. We feel sure that dialysers optimised for increased internal transport that aid the disruption of the laminar boundary layer will be a significant aid to push-pull HDF in the future and would be an interesting suite of experiments to perform in the future.

---

## [Decision Letter · Decision Letter 1]

3 Feb 2020

Enhancement of solute clearance using pulsatile push-pull dialysate flow for the Quanta SC+: a novel clinic-to-home haemodialysis system

PONE-D-19-16719R1

Dear Dr. Buckberry,

We are pleased to inform you that your manuscript has been judged scientifically suitable for publication and will be formally accepted for publication once it complies with all outstanding technical requirements.

With kind regards,

Pasqual Barretti, Ph.D., MD

Academic Editor

PLOS ONE

Additional Editor Comments (optional):

I agree with the reviewers. I believe that the authors address all questions and the manuscript has improved a lot, being able to be published.

Reviewers' comments:

Reviewer's Responses to Questions

**Comments to the Author**

1. If the authors have adequately addressed your comments raised in a previous round of review and you feel that this manuscript is now acceptable for publication, you may indicate that here to bypass the “Comments to the Author” section, enter your conflict of interest statement in the “Confidential to Editor” section, and submit your "Accept" recommendation.

Reviewer #1: (No Response)

Reviewer #2: All comments have been addressed

2. Is the manuscript technically sound, and do the data support the conclusions?

Reviewer #1: (No Response)

Reviewer #2: Yes

3. Has the statistical analysis been performed appropriately and rigorously? 

Reviewer #1: (No Response)

Reviewer #2: Yes

4. Have the authors made all data underlying the findings in their manuscript fully available?

Reviewer #1: (No Response)

Reviewer #2: Yes

5. Is the manuscript presented in an intelligible fashion and written in standard English?

Reviewer #1: (No Response)

Reviewer #2: Yes

6. Review Comments to the Author

Reviewer #1: (No Response)

Reviewer #2: Thank you for having addressed my concerns, even if sometimes you were not able to provide the precise or adequate answer. It reads better and more scientifically exact now.

7. PLOS authors have the option to publish the peer review history of their article (what does this mean?). If published, this will include your full peer review and any attached files.

Reviewer #1: No

Reviewer #2: No

---

## [Editor Report · Acceptance letter]

12 Feb 2020

PONE-D-19-16719R1 

Enhancement of solute clearance using pulsatile push-pull dialysate flow for the Quanta SC+: a novel clinic-to-home haemodialysis system 

Dear Dr. Buckberry:

I am pleased to inform you that your manuscript has been deemed suitable for publication in PLOS ONE. Congratulations! Your manuscript is now with our production department. 

With kind regards,

on behalf of

Dr. Pasqual Barretti 

Academic Editor

PLOS ONE